# Electroencephalogram-Based Evaluation of Impaired Sedation in Patients with Moderate to Severe COVID-19 ARDS

**DOI:** 10.3390/jcm11123494

**Published:** 2022-06-17

**Authors:** Armin Niklas Flinspach, Sebastian Zinn, Kai Zacharowski, Ümniye Balaban, Eva Herrmann, Elisabeth Hannah Adam

**Affiliations:** 1Department of Anaesthesiology, Intensive Care Medicine and Pain Therapy, University Hospital Frankfurt, Goethe-University Frankfurt, Theodor-Stern Kai 7, 60590 Frankfurt, Germany; sebastian.zinn@kgu.de (S.Z.); kai.zacharowski@kgu.de (K.Z.); elisabeth.adam@kgu.de (E.H.A.); 2Department of Biostatistics and Mathematical Modelling, Goethe-University Frankfurt, Theodor-Stern Kai 7, 60590 Frankfurt, Germany; balaban@med.uni-frankfurt.de (Ü.B.); herrmann@med.uni-frankfurt.de (E.H.)

**Keywords:** critical care, hypnotics and sedatives, acute respiratory distress syndrome, severe acute respiratory syndrome coronavirus 2, electroencephalogram, index of consciousness-view monitor, neurophysiological monitoring

## Abstract

The sedation management of patients with severe COVID-19 is challenging. Processed electroencephalography (pEEG) has already been used for sedation management before COVID-19 in critical care, but its applicability in COVID-19 has not yet been investigated. We performed this prospective observational study to evaluate whether the patient sedation index (PSI) obtained via pEEG may adequately reflect sedation in ventilated COVID-19 patients. Statistical analysis was performed by linear regression analysis with mixed effects. We included data from 49 consecutive patients. None of the patients received neuromuscular blocking agents by the time of the measurement. The mean value of the PSI was 20 (±23). The suppression rate was determined to be 14% (±24%). A deep sedation equivalent to the Richmond Agitation and Sedation Scale of −3 to −4 (correlation expected PSI 25–50) in bedside examination was noted in 79.4% of the recordings. Linear regression analysis revealed a significant correlation between the sedative dosages of propofol, midazolam, clonidine, and sufentanil (*p* < 0.01) and the sedation index. Our results showed a distinct discrepancy between the RASS and the determined PSI. However, it remains unclear to what extent any discrepancy is due to the electrophysiological effects of neuroinflammation in terms of pEEG alteration, to the misinterpretation of spinal or vegetative reflexes during bedside evaluation, or to other causes.

## 1. Introduction

The severe course of the coronavirus 2019 (COVID-19) disease continues to cause critical oxygenation disorders with the need for mechanical ventilation and consecutive sedation, especially among unvaccinated individuals.

The treatment of severe COVID-19-associated acute respiratory distress syndrome (CARDS) usually requires deep sedation for prone positioning and invasive ventilation [1]. However, this sedation management and treatment proves to be challenging in CARDS patients due to a significant increase in the need for sedation, whose causes remain so far unknown [2,3,4,5]. The underlying pathological factors of this aggravation are still largely unknown [6]. Thus, it has not yet been investigated to what extent the necessary high sedative dosages and combination applications are actually due to inadequate sedation levels [7], or whether it is a misinterpretation of the depth of sedation when vegetative and spinal reflexes occur. Besides that, an increased rate of agitation and encephalopathy has already been shown [8,9]. To what extent the required deep sedation of critically ill CARDS patients and its measurement by pEEG contributes to, e.g., the delirium of CARDS patients is still unclear [10,11].

In daily clinical practice, there is no objective measure for assessing the depth of sedation that is sufficient and appropriate for all situations. An adjustment of sedation is often based on bedside observations by the attending physician, such as movements in terms of their subjective experience. At present, the clinical determination of adequate sedation depth is predominantly done by assessing the reliable Richmond Agitation and Sedation Scale (RASS), the occurrence of vegetative agitation (tachycardia, hypertension, sweating, tachypnea, and tears in the eyes not otherwise explained), and ventilator synchrony. For the various treatments required in patients with CARDS, such as prone positioning or veno-venous extracorporeal membrane oxygenation (VV-ECMO), deep sedation (RASS −3/−4) is recommended [12,13]. In line with the institutional preferences and current practice, deep sedation is maintained using a variety of substances, such as benzodiazepines (e.g., midazolam), central alpha-2 agonists (e.g., dexmedetomidine), esketamine, and propofol, as well as volatile anesthetics (e.g., isoflurane) and opioids (e.g., sufentanil). Along with sedation, non-depolarizing neuromuscular blockers (NMBA) (e.g., rocuronium) and opioids (e.g., fentanyl) are often used in the initial phase [5,14]. The RASS is primarily assessed and measured by the duration of eye-opening or eye movement [15]. However, the ability to open the eyes is affected by prone positioning, the use of NMBAs, or progressive muscular weakness in terms of intensive care unit-acquired weakness (ICUAW) [16]. These conditions are common in critically ill CARDS patients, which may cause a serious impairment of the sedation assessment [17,18,19].

So far, alternatives to RASS have been evaluated in assessing sedation depth or cerebral activity by using encephalography (EEG) in a small number of patients [20,21]. The assessment of an adequate sedation depth using EEG in intensive care units has already been demonstrated [22,23]. However, the conventional bedside recording and interpretation of raw EEG waveforms requires a considerable technical effort and extensive staff training. For this reason, less complex measurement techniques have been introduced in recent years that can be handled by non-neurological professionals [24]. The application of processed electroencephalography (pEEG) has previously been demonstrated for the detection of burst suppression as well as for the adjustment of the depth of anesthesia in intraoperative and intensive care settings [25,26].

Devices that generate a simple bedside numerical index that can be understood by non-experts in the field of encephalography are already available. Such an EEG-based sedation index, such as the patient status index (PSI), is calculated via an algorithm from a high-resolution multi-channel electroencephalogram after artifact suppression [27]. Among others, the PSI was developed specifically for critical care to monitor the sedation of patients [28,29]. The EEG scores obtained by algorithms demonstrated comparability to clinical sedation scores on the Observer’s Assessment of Alertness/Sedation Scale (OAA/S) or the RASS [30,31]. The algorithm relies on EEG power, frequency, and phase information from anterior-posterior relationships of the brain as well as on the coherence between bilateral brain regions [29]. This prospective study aims to evaluate sedation monitoring in critically ill COVID-19 patients without the prior suspicion of a neurological disorder. Since the data on the utility of processed EEG data in the sedation of CARDS patients are largely lacking, we conducted an exploratory study. We describe PSI values under the influence of different sedative drugs and the bedside assessment of RASS.

## 2. Methods

This is a prospective observational study conducted at the University Hospital of Frankfurt which has been approved by the institutional ethics board of the University of Frankfurt (#2021-7). Due to the prospective nature of the study, an informed consent form was obtained from each individual patient or their legal representative. The study was registered prior to the study at clinicaltrails.gov (NCT 04815109) before the first patient enrollment. This manuscript adheres to the applicable CONSORT guidelines and the Declaration of Helsinki and its later amendments.

## 3. Patient Population

The patient recruitment took place between May 2021 and December 2021 among critically ill COVID-19 patients.

The inclusion criteria were: SARS-CoV-2 virus-associated ARDS with the need for mechanical ventilation and corresponding sedation; a patient age > 18 years; and the written informed consent of the study participant or their legal representative. The exclusion criteria were: patients with pre-existing severe cerebral impairment as well as dysfunction; minors; patients > 80 years of age; and the withdrawal of prior consent.

The treatment of patients was performed according to the current recommendations for critically ill CARDS patients [32,33,34]. With regard to analgesia, the continuous intravenous administration of a strong opioid (e.g., sufentanil) in combination with the continuous intravenous administration of a sedative was used. In line with the existing standards of our ARDS center and the ABCDE guidelines, if analgesia was necessary and sufficient, escalation with further sedatives was performed according to the decision of the attending physician [35]. Neuromuscular blocking agents (NMBA) were used in exceptional cases only and not as standard to allow for the spontaneous ventilation and optimization of oxygenation [36,37]. The included patients received mechanical ventilation with an Elisa 800 (Löwenstein Medical, Bad Ems, Germany) or Hamilton G5 (Hamilton Medical, Bonaduz, Switzerland) ventilator. A definite patient history with the identification of possible cofounders such as alcohol or substance abuse was not obtainable. However, a differentiated analysis of past medications, the available medical history, and the laboratory results did not reveal any cases suspicious for such abuse. EEG measurements were performed during constant continuous sedation, excluding previous bolus applications. To determine the adequacy of sedation, the reliable RASS and a bedside examination were performed by the attending physician, who assessed particular signs of stress and the occurrence of vegetative agitation [38]. According to current recommendations, an RASS of −1 to −2 was targeted in the standard of care of mechanically ventilated patients, while an RASS of −3 to −4 was targeted in patients receiving prone positioning and ECMO therapy for adequate psycho-vegetative dissociation [39,40].

## 4. Outcome Parameters

Primary Outcome Measure:Processed encephalography measurement

Secondary Outcome Measures:Dosage of clonidine [µg/kg/h]Dosage of propofol and midazolam [mg/kg/h]Dosage of esketamine [mg/kg/h]Dosage of sufentanil [µg/kg/h]

## 5. EEG Data Acquisition

Due to hygiene-related and practical reasons, the encephalography was performed with a device that is easy to handle and clean. The Masimo–Root–SedLine^®^ system (Masimo Corporation, Irvine, CA, USA) meets these requirements and is able to generate a Patient State Index (PSI) as an EEG sedation depth score. A superficial EEG was recorded accordingly using a Masimo–Root–SEDLine™ system. Via six frontal electrodes—Fp1, Fp2, F7, and F8—according to the international 10–20 system, the electrical activity of the brain was recorded, with the reference and ground electrode placed in the middle of the forehead (around Fz). The SEDLine monitor processes the EEG information to calculate the PSI. The adhesive electrode was attached according to the manufacturer’s instructions. After the stabilization of the measured parameters, a 30 min data collection period followed. To achieve a stable sedative blood plasma level, no changes in the continuous sedative dosage were made in the time interval of at least two hours before the measurement. No manipulations were performed with the patient during the EEG measurement in order to achieve appropriate shielding. The RASS was assessed at the beginning and end of the measurement. The electrodes were routinely checked to keep impedances < 5 kiloohms in order to ensure a high signal quality. The SEDLine™ system provides a Patient State Index (PSI) value using an algorithm based on the Fast Fourier Theorem to analyze the raw EEG [41]. The value is determined by real-time computation to represent the depth of sedation. The real-time computations of the PSI index are considered adequately competent to predict the depth of sedation with various anesthetics regimens. A second 30 min EEG recording was performed the following day with the same technique (processed EEG recording). The EEG measurements and subsequent data extractions were performed under the same unmodified test conditions to avoid any influence on the data [42].

## 6. Data Collection

Electroencephalographic data were obtained from the Masimo–Root–SedLine^®^ system via The Masimo Instrument Configuration Tool^TM^ V.1.2.4.5. (Masimo Corporation, Irvine, CA, USA) for further analysis. The clinical data were recorded using an intensive care unit patient data management system (PDMS; Metavision 5.4, iMDsoft, Tel Aviv, Israel). We recorded demographic data, sedative and analgetic dosages, the clinical satisfaction of sedation levels, RASS, positioning therapy, VV-ECMO therapy, fluid balance, and outcomes (death or discharge).

## 7. Statistical Analysis

Prior power calculation was conducted under the guidance of the Institute of Biostatistics and mathematical modeling. The primary statistical objective was to test the correlation influence of the EEG as well as the calculated Patient State Index on the patients’ sedation level via linear regression analysis. The case number estimation with a significance level of alpha = 5% and an assumed correlation of 0.5 resulted in the required number of patients of *n* = 46 to reach a power of 95%. The data with the continuous scale are represented as the median (interquartile range, IQR), and the data with the categorical scale are presented as frequencies and percentages.

Additionally, the PSI, sedative dosages, and RASS values were analyzed using linear regression models with a mixed effect and using a correlation matrix with an autoregressive process calculated from a subset of markers selected for their ability to infer ancestry. In addition, for the models, data adjustments were made with respect to variance components.

All of the statistical tests were two-tailed, and the results with *p* ≤ 0.05 were considered statistically significant. All of the calculations/analyses were performed with SPSS (IBM Corp., Version 26, Chicago, IL, USA) or R for Statistical Computing (The R Foundation, Version 4.0, Vienna, Austria). The packages ‘MASS’ and ‘nlme’ were used [43,44].

## 8. Results

The analysis of the study was based on the EEG data of 49 patients measured twice for 30 min. In one patient, only a single measurement could be performed due to sudden death (see Figure 1). A mean (±SD) of 1864 (±585) values corresponding to 63 min of data recording could be obtained from each patient. The demographics of the patients investigated are shown in Table 1.

The initial pEEG measurement was performed on a mean of 8.5 (±5.8) days after the necessity of endotracheal intubation with mechanical ventilation and subsequent sedation. At the time of the measurement, none of the patients were in the effect of an NMBA. Muscular activity was detected in the electromyogram (EMG) in 20.2% of the observation period. Among the EMG values obtained, we found a median muscular activity of 17% (IQR 40%). The studied patients had a positive fluid balance of a median of 2.39 (IQR 5.6) liters within the previous intensive care treatment. The PSI values calculated from the pEEG measurements revealed a median value of 20 (IQR 8). The distribution of the individual PSI values is presented in the patient-associated plots in Figure 2. The detected suppression rate (SR) was 14 (±24) at the median; for the 95% spectral edge frequency (SEF95), we found 8.1 (IQR 8.1) on the right and 7.7 (IQR 7.1) on the left parietofrontal forehead.

During 79.4% of the pEEG recordings, deep sedation (corresponding to RASS −3/−4) was determined in the bedside examination. A differentiated distribution of the determined RASS levels can be found in Figure 3A. Furthermore, spontaneous coughing, swallowing, gagging, and pressing with the partial difficulty of the respirator synchronization were observed during the measurements. The linear regression analysis revealed a significant correlation between the sedative dosages of propofol, midazolam, clonidine, esketamine, sevoflurane, and the opioid sufentanil (*p* < 0.0001) and the PSI (Table 2, Figure 4A–E) Furthermore, when comparing the recorded RASS under sedation (primary evaluation of eye opening), a significant correlation was found with the measured PSI (*p* < 0.0001) The Bland–Altmann diagram (Figure 4F) of this correlation shows a dependence strongly influenced by the PSI value, resulting from the non-normalization of the RASS and PSI values. The analysis also considered the prone positioning and changes in the sedation therapy as potential confounders: four patients were transferred from the prone position to the supine position between both of the pEEG measurements performed, and no change in VV-ECMO therapy was observed.

The doses of the sedatives and opioids continuously applied during the pEEG measurements are shown in Figure 3B. The sedation depth required was achieved by 85.6% with the use of multiple sedative substances. During fourteen pEEG measurements, sedation was achieved with a single sedative (propofol, clonidine, or sevoflurane). There was no correlation between the fluid balancing or oxygenation level and the corresponding PSI or SEF 95 determined.

## 9. Discussion

Processed EEG is becoming more and more popular for the management of deep sedation in the context of the intensive care treatment of critically ill CARDS patients. The system we use, Masimo SedLine, is one of many devices available for assessing pEEG by reproducing SR, SEF, EMG, and PSI [23,27,28]. The extent to which the previously demonstrated utility of PSI for sedation management guidance in critically ill non-COVID patients also applies to a cohort of patients with impaired sedation and concomitant neuroinflammation, such as patients with CARDS, remains unclear. To achieve deep sedation, PSI values of 25–50 were expected, but our results are not consistent with this assumption. The majority of subjects showed substantially lower values, indicating a deep coma [27]. In this context, burst suppression with correspondingly high SR values would be expected, but the SR we obtained did not confirm this relationship [46,47].

The PSI we recorded is more consistent with the observation of a frequent alpha coma, as described previously in CARDS patients [48]. However, as shown in Figure 2, the cohort we analyzed showed an unusually high PSI in twelve sampling intervals, which seems to imply that the patient was awake. Interestingly only subject 1 showed high PSI values in both measurements. Moreover, the corresponding patients were in clinically anticipated deep sedation, and most were under combined continuous sedative treatment. This observation further challenges the interpretability of the PSI in CARDS. Both attending physicians and nurses repeatedly expressed the marked discrepancy between the clinical observation of the sedation depth of the patients, who displayed a motion response upon physical contact, and the PSI determined by using pEEG. Thus, subject 24 could be extubated without problems immediately after the completion of the pEEG measurement, due to advanced respirator weaning with a PSI of 20 (see Figure 2), without any change in sedative medication and with an adequate neurological response in terms of sufficient vigilance and cooperation.

The majority of patients demonstrated a PSI of less than 25, although an RASS of −1 to −3 was repeatedly noted. This was unexpected considering that a PSI of 25–50 has been demonstrated to be sufficient during general anesthesia and is more likely to correspond to an RASS of −5 [28,29,49].

We found a significant correlation between the PSI and the RASS, which is expected and not surprising given that both methods are used for sedation depth measurement. The arousability by stimulation in the sedated ICU patients correlates with the depth of sedation, as measured with the RASS and expressed via the PSI [50]. Interestingly, a closer look reveals inconsistencies: the RASS does not adequately reflect the clinically observed depth of sedation. The PSI values obtained indicated a much lower RASS (e.g., −5) than that determined in the clinical examination (−3/−4) [50]. Regarding the sedation of critically ill patients, there exists no consensus on the method to be used. In addition to the assessment by nurses, scores such as the RASS, the Ramsay and modified scales, and the Glasgow Coma Scale are used in clinical practice [51]. However, the RASS can only represent one component of the assessment of deep sedation due to existing limitations [52]. Processed EEGs are more promising, especially with regard to the maintenance and assessment of moderate and deep sedation. However, the transferability of the bedside sedation assessment and the scales evaluated to the indices obtained by pEEG is still not yet clear [41]. The applicability of pEEG to the special population of ARDS patients has only been demonstrated occasionally [53]. In addition, studies demonstrating a good transferability of PSI to bedside sedation scores are lacking to date. Therefore, the transferability of a pEEG application to non-CARDS patients is impeded by the significantly increased sedation requirements of CARDS patients and represents an important limitation [2,3]. As expected, we were able to demonstrate a significant correlation for the GABA_A_ receptor-active sedatives propofol and midazolam, the central α_2_-agonist clonidine, the opioid sufentanil, and the volatile substance sevoflurane. Such a significant correlation could also be shown for esketamine, which, however, partly contradicts the previously described increase in pEEG indices under esketamine application and seems to vary depending on the dose regimens [54,55,56]. Esketamine shows some peculiarities in the EEG patterns under sedation that could explain the outliers of the index values [57]. This observation is most likely due to the consistent combination with other sedatives. For example, the patients described in Figure 2, in whom PSI outliers were documented, did not receive esketamine.

Even though the majority of devices available for recording pEEG for sedation control are based on Fast Fourier Analysis, the comparison is limited. This is mainly due to the proprietary algorithms of the manufacturer. The influence of SR, SEF, artifact analyses, and EMG remains rather difficult to assess—especially because various studies revealed severe limitations of the parameters EMG and SR on sedation indices [46,58]. Therefore, we decided to perform the pEEG measurements without muscle relaxation. Nevertheless, only a small percentage of the measurements showed muscular activity in the sense of a derivable EMG activity. Whereas the statistical correlation of age and determined pEEG could already be shown in non-COVID patients, we failed to show such a relationship [45,59].

The ability of the SARS-CoV-2 virus to affect the central nervous system was recognized early by the occurrence of anosmia and ageusia [60]. Due to the growing knowledge about the virus, attention is increasingly focusing on the neuroinflammation frequently observed in critically ill patients [61]. Hence, it remains to be discussed to what extent neuronal impairment due to neuroinflammatory processes influences the algorithms of pEEG and affects the validity of the studies performed in non-COVID-19 patients. This would be a possible explanation for the unusually low PSI values we found. However, it is important to consider that there may be a repeated misinterpretation of the validated sedation scores such as SAS and RASS. Depending on the patient’s behavior, a motoric reaction may be interpreted due to the risk of mistaking spinal reflexes such as swallowing, coughing (sometimes with rearing up in bed), and pressing. Thus, the evaluation of the required deep sedation in severe COVID-19 patients remains difficult, even with the help of EEG [20]. Future analyses clarifying the effects of neuroinflammation may be warranted by using unprocessed EEG data.

Our study was conducted at a single center using only one device, without comparison to, for example, raw EEG or the BIS model. This is especially due to the time-consuming procedure of connection and measurement in times of increasing scarcity of human resources [62]. Further limitations are that the present study is a monocentric study and that no continuous pEEG measurement was performed.

## 10. Conclusions

Our results revealed a distinct discrepancy between the bedside patient evaluations determined by the RASS and the PSI obtained by pEEG. The causes of this discrepancy cannot be conclusively determined. Possible causes may be vegetative and spinal reflexes as well as the unsuitability of the pEEG application in neuroinflammatory processes in the context of COVID-19 infection. However, it remains to be clarified whether this discrepancy is due to electrophysiological effects of neuroinflammation in the sense of a pEEG modification or a misinterpretation of reflexes during the bedside evaluation of the RASS.

## Figures and Tables

**Figure 1 jcm-11-03494-f001:**
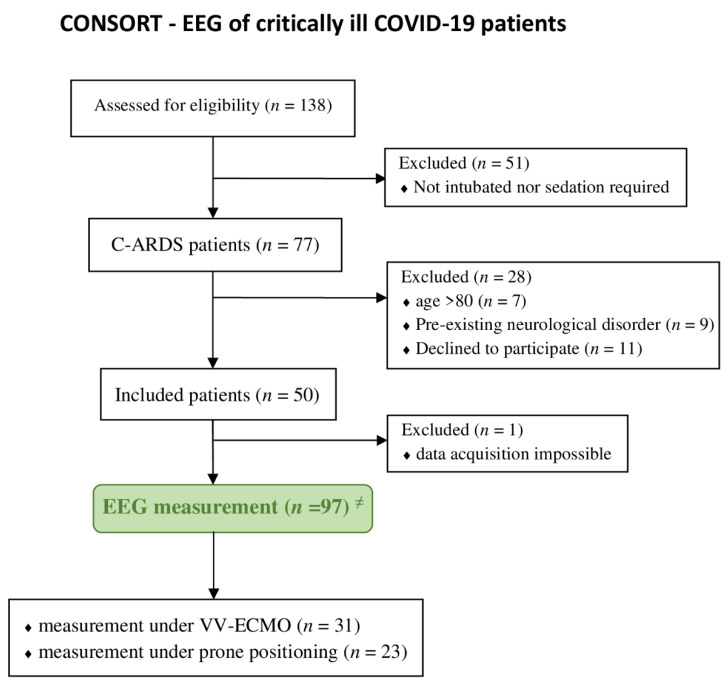
Flowchart of patients included in the study (according to the CONSORT criteria). Abbreviations: CARDS, coronavirus-associated acute respiratory distress syndrome; EEG, electroencephalography; VV-ECMO, veno-venous extracorporeal membrane oxygenation. ^≠^ Each patient received two EEG recordings that were 30 min in duration, and one patient died before the second 2 EEG measurement.

**Figure 2 jcm-11-03494-f002:**
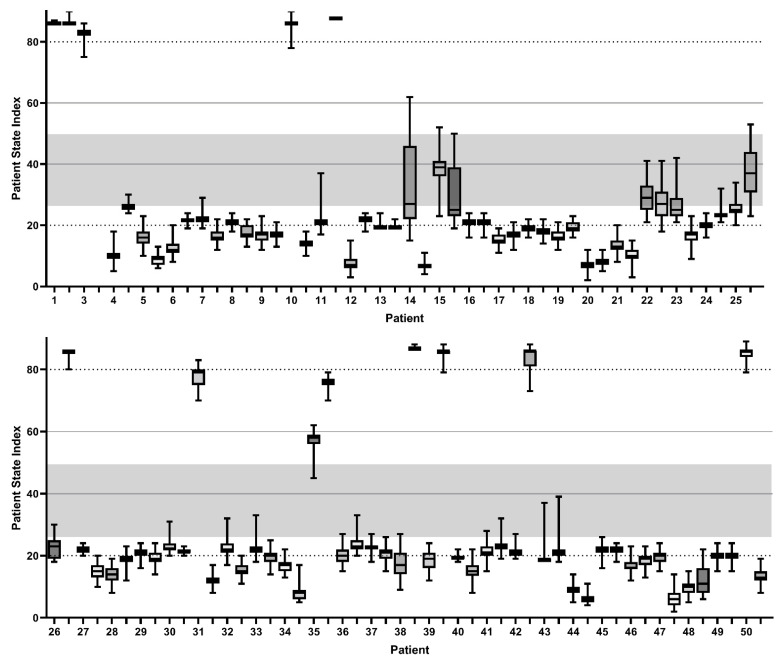
Patient State Index values of the patient side measurement. Box-and-whiskers plots of each measuring interval of a 30 min duration. Separate representation of the two performed individual pEEG recordings for each patient. Patient State Index range of 25–50, corresponding to the expected value of sufficient sedation depth [23,45].

**Figure 3 jcm-11-03494-f003:**
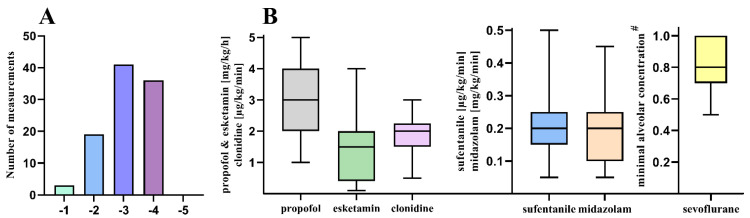
Sedation level and sedatives applied. Analysis of the sedation levels and box-and-whiskers plots of the dosages of the administered sedatives. (**A**). Distribution of bedside-determined Richmond Agitation and Sedation Scale level at the beginning of the pEEG measurement. (**B**). Dosages of continuous intravenous sedatives including the volatile sedative sevoflurane and the opioid sufentanil. Abbreviations: µg, microgram; mg, milligram; kg, kilogram bodyweight; min, minute; h, hour. ***^#^*** minimal alveolar concentration (MAC) as weight-age and sex-adjusted MAC_50_.

**Figure 4 jcm-11-03494-f004:**
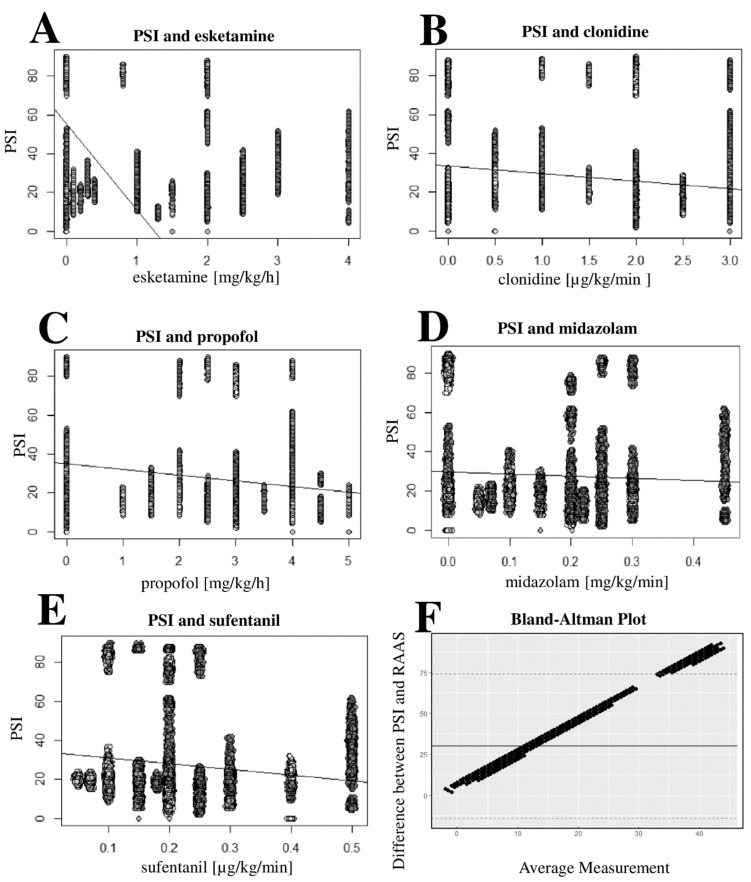
Scatter plots of PSI and sedative dosages; Bland–Altmann plot of PSI and RASS. Patient State Index scatter plots showing esketamine (**A**), clonidine (**B**), propofol (**C**), and midazolam (**D**), as well as the opioid sufentanil (**E**). Bland–Altmann plot of the PSI and the sedation Score RASS (**F**). Abbreviations: µg, microgram; mg, milligram; kg, kilogram bodyweight; min, minute; h, hour; RASS, Richmond Agitation and Sedation Scale; PSI, Patient State Index.

**Table 1 jcm-11-03494-t001:** Clinical characteristics of C-ARDS patients.

Characteristics	Patients Included *n* = 49	
age, y	52	(18)
sex, male	38	[77.6%]
bodyweight, kg	90.0	(±19.8)
BMI ^≠^	29.4	(7.1)
SAPS II ^≠^	43.9	(29)
p_a_O_2_x FiO_2_^−1 ≠^	98.0	(81)
VV-ECMO treatment	*n* = 31	[63.3%]
mortality	*n* = 22	[44.9%]
coronary artery disease ^≠^	*n* = 10	[20.4%]
pulmonary disease ^≠^	*n* = 8	[16.3%]
diabetes ^≠^	*n* = 18	[37.7%]
arterial hypertonus ^≠^	*n* = 30	[61.2%]
chronic kidney disease ^≠^	*n* = 10	[20.4%]

Data are presented as the mean (IQR) or as the patient number [percentage], where applicable. Abbreviations: BMI, Body mass index; kg, kilogram; SAPS II, Simplified Acute Physiology Score II; SD, Standard deviation; VV-ECMO, veno-venous extracorporeal membrane oxygenation; y, years. ^≠^ determined on admission.

**Table 2 jcm-11-03494-t002:** Results of the univariate linear regression analysis. Results of the multivariate linear regression analysis regarding the determined Patient State Index and sedation dosages administered. Abbreviations: RASS, Richmond Agitation and Sedation Scale; µg, microgram; mg, milligram; kg, kilogram bodyweight; min, minute; h, hour; MAC, minima alveolar concentration; IQR, interquartile range; CI, confidence interval.

Variable	Estimate	Standard Error	*p*-Value	CI 95%	Mean Dosage [IQR]
propofol	8.9	0.18	<0.0001	(8.5;9.3)	3.0 [2.0] mg/kg/h
midazolam	−192.12	2.97	<0.0001	(−197.9; −186.3)	0.2 [0.1] mg/kg/min
clonidine	−32.11	0.42	<0.0001	(−32.9; −31.3)	2.0 [0.5] µg/kg/min
sufentanil	−85.74	1.59	<0.0001	(−88.9; −82.6)	0.2 [0.1] µg/kg/min
esketamine	60.09	1.54	<0.0001	(57.1; 63.1)	1.5 [1.15] mg/kg/h
RASS	12.5	0.18	<0.0001	(12.1; 12.9)	

## Data Availability

The data cannot be shared publicly. The datasets generated and/or analyzed during the current study are not publicly available due to national data protection laws but are available upon reasonable request from the corresponding author or via the data protection officer of the University Hospital of Frankfurt (Datenschutz@kgu.de (www.kgu.de, accessed on 19 March 2022)).

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
