# Peer review of "Electroencephalogram-Based Evaluation of Impaired Sedation in Patients with Moderate to Severe COVID-19 ARDS"

_jcm, 2022, doi:10.3390/jcm11123494_

Round 1
Reviewer 1 Report
The authors have created a pragmatic approach to evaluating the increased requirements of sedation for COVID-19 related ARDS. While a variety of alterations to the study design, such as continuous and simultaneous raw EEG to compare to the pEEG, could have improved the study, the authors provided practical approach to evaluating this problem. Please see my comments below.
-Abstract: Including the expected PSI value for the RASS of _3 to -4 would provide better context to the reader for the discrepancy
-Line 43: The term "level of consciousness with content" is unfamiliar and unclear to me.
-Line 67: I suspect this is a typo where "fusing" is written instead of "using", though fusing EEG has been used as a term to fuse EEG and fMRI.
-Line 77: Are there known relationships of PSI values and RASS that can be cited in this paragraph? Alternatively, this could be included in the methods where the SEDLine is described.
-Line 95: Were potential cofounders such as history of substance abuse, alcoholism, chronic pain, or any other confounders considered or controlled for?
-Line 141: Was a simultaneous raw EEG and pEEG considered? This would have reduced the potential for confounders or changes in patient status the following day. Additionally, it is not apparently clear to me if the second reading the following day was the same protocol using pEEG or if it was a raw EEG. As I read the remainder of the manuscript, I understand it is a second pEEG. Please try and clarify this in the methods that the second EEG is a pEEG.
-Table 1: SAPS II, PaO2/FiO2 were not variables discussed in the methods section yet they are here in the results. Please add these and any other variables collected to the methods.
-Results: do you have any information on what differences existed between the two data points collected? For example, what percentage of patient were proned for one recording and supine for the second? Other aspects to consider are changes in VV ECMO or sedation medications being used.
-line 195: What does the "median of 17" refer to? Please provide context/units.
Line 240: Consider combining these three sentences to flow better: "The PSI levels we determined suggest unusually deep sedation in CARDS patients. To achieve deep sedation, PSI values of 25-50 were expected. Our results do not follow this assumption" Such as "To achieve deep sedation, PSI values of 25-50 were expected, yet our results do not follow this assumption as the PSI levels we determined suggest unusually deep levels of sedation is necessary in CARDS patients"
Reviewer 2 Report
Flinsbach et al. assessed the use of processed EEG in COVID-19 patients with moderate to severe ARDS. The topic is interesting, as COVID-19 patients need high doses of anesthetics for adequate sedation and the reason is still unknown. However, the reporting of the results is incomplete. It seems the reporting of the primary endpoint, the association of processed EEG (PSI) and sedation level (RASS), is missing in the results section. I, therefore, recommend reconsideration after major revisions. Please find more detailed comments below.
Introduction
L46-47: These two sentences do not follow your previous argumentation. Furthermore, they suggest you are investigated the association between sedation level and delirium. Please connect your thoughts and adapt this passage accordingly.
L54: I think it should be „is induced“.
L57: To mention the use of NMBAs for the induction of deep sedation is inadequate here. Please revise this statement.
L59-65: These three “However” sentences are not connected, and the final sentence does not clearly follow the argumentation of the paragraph. I think the statement on the careful use of NMBAs is not needed here, as this is kind of breaking down your argumentation. Additionally, please further clarify why it is particularly difficult to measure sedation depths in COVID-19 patients to clearly justify why this investigation was needed in these patients.
Please state the primary hypothesis within the last paragraph of the introduction. If this is an explorative study without an a priori hypothesis, then please state so.
Methods
You state that this manuscript adheres to the CONSORT guidelines which usually applies to clinical (interventional) trials. However, the presented research is purely observational and most likely categorized as a cross-sectional study. Thus, the STROBE guidelines should be used to ensure the quality of reporting. Please use the applicable STROBE checklist and make sure to revise the manuscript accordingly.
Please insert a separate subchapter to clearly describe primary and secondary outcomes/endpoints of the study.
Statistics
L155-157 “influence” should be replaced by correlation. I guess the primary objective was also assessed by linear mixed effects models to account for repeated measures, as stated in L163. If so, please also clarify here.
L159 Medians are usually presented along with interquartile ranges, and means along with standard deviation. Why did you pick the combination of median and SD?
L156 The English “respectively” is not used the same way as the German “bzw.”. Please revise this passage.
Results
The primary endpoint of this study seems to be the correlation between PSI and sedation level (RASS), as the study was powered for his endpoint. However, this endpoint is not reported in the manuscript. Please present all your results.
Presenting only the p-value for a correlation is needless. You should provide the estimates of your linear mixed model including 95% confidence intervals to enable the reader to assess the strength of the correlation. Furthermore, scatter plots including the regression line of the calculated model should be included to present the primary endpoint.
Discussion
You report that about 80% of the patients were deeply sedated at RASS -3/-4 and most of the patients had low PSI values. So where is the discrepancy between RASS and PSI?
When you include the correlation of PSI and RASS (primary endpoint), please compare your findings to other studies.
You discuss neuroinflammation as a potential reason for surprisingly low PSI values. Please consider consulting a neurologist to assess the raw EEG for signs of neuroinflammation to give this statement a scientific basement.
Did the patients with high PSI values receive ketamine? Does this explain at least some of the outliers?
“Our results revealed a distinct discrepancy between the bedside patient evaluations determined by RASS and the PSI obtained by pEEG.” On which basis did you come to this conclusion? You don’t present the association of RASS and PSI.
Round 2
Reviewer 2 Report
The manuscript has much improved. However, due to the inconsistent reporting of aims, outcomes and endpoints, as well as the quality of the reporting of the results, the manuscript still needs major revisions. Aside revisions of the content, please make sure that the manuscript has undergone lingual revision, as some passages are still hard to read.
Introduction
The introduction has much improved. The definition of the study aim is still not precise. What do you mean by an exploratory analysis of pEEG patterns? Please make sure that the aim is consistent with your defined outcomes/endpoints and later reporting.
Methods
The described outcomes differ from the ones registered at clinicaltrial.gov. They should stay unchanged, otherwise prospective registration is senseless. For clarification, outcome is the clinical measure you would like to assess (e.g. pEEG values, RASS), endpoint is the statistical measure you are aiming to assess (e.g. correlation). Please use the terms accordingly.
The current definition of the primary endpoint sounds like the aim of the study, but not like an outcome: "Comparability between PSI and clinically determined sedation depth including sedation administered.” Furthermore, how is comparability defined? What is the measure of sedation depth (RASS)? What do you mean by sedation administered? Please be precise.
Sorry, but you can't power the study for the correlation between pEEG (PSI) and sedation depth (RASS) and then say it’s not the primary aim to assess this correlation.
Results
Again, significance for correlation is meaningless. There are still no scatter plots reported. The described appendix with scatter plots is completely missing, making it impossible to assess the variability of the correlation. Furthermore, for the comparison of two measurement methods like PSI and RASS, Bland-Altmann plots are usually reported. These plots will enable you and the reader to assess the deviations between the two measures for sedation depth. I suggest including them.
Table 2: The standard error is somewhat hard to interpret. Please report 95% confidence intervals, as suggested in my previous review. The legend describes “univariate linear regression analysis” as the statistic used but I thought you used linear mixed models to account for within subject correlation by repeated measurements. The use of simple linear regression would be inadequate here. This is a huge difference. Please be precise and consistent in the reporting throughout your manuscript.
The authors state: “Not further quantified, both attending physicians and nurses repeatedly expressed the marked discrepancy between the clinical observation of the sedation depth of the patients, who displayed motion response upon physical contact, and the PSI determined by using pEEG.” This is describing subjective feelings. If motion response was not quantified, this should not be part of evidence-based reports.
Discussion
“The extent to which the previously demonstrated correlation of PSI in critically ill non-COVID patients is also applicable in a cohort of patients with impaired sedation and concomitant neuroinflammation, such as patients with CARDS, remains unclear.” - The correlation of PSI with what measure?
I guess, there are plenty of studies that assessed the correlation of PSI with clinically determined sedation depth or sedative medication. How do your results in CARDS patients compare to other patient populations? Is there a difference?
L299-304: These statements rely on unquantified measures and reflect a subjective feeling of the treating medical professionals. This should not be overinterpreted.
L305-320: The authors mention that RASS is not predicting coughing, swallowing, gagging, and spontaneous movement. However, this was not the research question of this study.
In general, it seems the authors are mixing up two research questions. First is the assessment of the correlation of PSI with established clinical measures of sedation depth like RASS. Second is the assessment of the predictive value of PSI and RASS for physical responses. The first research question can be adequately addressed by the presented data, whereas data such as asynchrony on the respirator, coughing and movements needed to address the second research question is missing. The discussion currently focusses too much on the second – unassessed – question. The authors should focus on answering and discussing the questions that are assessable by the obtained data.
